# Is Running Power a Useful Metric? Quantifying Training Intensity and Aerobic Fitness Using Stryd Running Power Near the Maximal Lactate Steady State

**DOI:** 10.3390/s23218729

**Published:** 2023-10-26

**Authors:** Cody R. van Rassel, Oluwatimilehin O. Ajayi, Kate M. Sales, James K. Griffiths, Jared R. Fletcher, W. Brent Edwards, Martin J. MacInnis

**Affiliations:** 1Faculty of Kinesiology, University of Calgary, Calgary, AB T2N 1N4, Canada; crvanras@ucalgary.ca (C.R.v.R.);; 2Department of Health and Physical Education, Mount Royal University, Calgary, AB T3E 6K6, Canada

**Keywords:** wearable technology, running economy, critical intensity, human performance, inertial measurement unit, treadmill

## Abstract

We sought to determine the utility of Stryd, a commercially available inertial measurement unit, to quantify running intensity and aerobic fitness. Fifteen (eight male, seven female) runners (age = 30.2 [4.3] years; $\overset{\cdot }{V}$O_2_max = 54.5 [6.5] ml·kg^−1^·min^−1^) performed moderate- and heavy-intensity step transitions, an incremental exercise test, and constant-speed running trials to establish the maximal lactate steady state (MLSS). Stryd running power stability, sensitivity, and reliability were evaluated near the MLSS. Stryd running power was also compared to running speed, $\overset{\cdot }{V}$O_2_, and metabolic power measures to estimate running mechanical efficiency (EFF) and to determine the efficacy of using Stryd to delineate exercise intensities, quantify aerobic fitness, and estimate running economy (RE). Stryd running power was strongly associated with $\overset{\cdot }{V}$O_2_ (R^2^ = 0.84; *p* < 0.001) and running speed at the MLSS (R^2^ = 0.91; *p* < 0.001). Stryd running power measures were strongly correlated with RE at the MLSS when combined with metabolic data (R^2^ = 0.79; *p* < 0.001) but not in isolation from the metabolic data (R^2^ = 0.08; *p* = 0.313). Measures of running EFF near the MLSS were not different across intensities (~21%; *p* > 0.05). In conclusion, although Stryd could not quantify RE in isolation, it provided a stable, sensitive, and reliable metric that can estimate aerobic fitness, delineate exercise intensities, and approximate the metabolic requirements of running near the MLSS.

## 1. Introduction

A consensus regarding an approach to evaluate mechanical running power output (PO) is lacking, resulting in a range of PO values for a given running speed, depending on the method [1,2]. During level running, the working muscles transfer energy to produce and absorb the forces needed to move body segments. As a result, there is no dissipative load external to the body that can be measured to quantify mechanical PO. Instead, running mechanical PO measurements may be derived from “external” or “internal” work perspectives by evaluating the centre of mass (CoM) or the body segments, respectively [1,2,3]. Such approaches require sophisticated laboratory assessments of joint kinetics and/or kinematics based on ground reaction force and motion-capture data. Several methodological challenges also limit the utility of running mechanical PO to approximate the metabolic work rate [1,3], and, in contrast to cycling, where there is a strong relationship between mechanical and metabolic PO [4,5], many factors complicate the relationship between mechanical and metabolic PO when running [6,7,8]. Nevertheless, a wearable running device that can quantify and monitor training intensity, analogous to a cycling power meter [9,10], would be useful to guide training and maximize running performance.

Several consumer technologies providing a running power metric have been developed [11,12]. These technologies derive a measurement of mechanical PO using estimates of ground reaction forces, CoM velocity, and/or vertical displacement from global positioning system (GPS) and/or inertial measurement unit (IMU) sensor data [11,12,13]. Previously, the Stryd running power device (a portable IMU), has provided the closest relationship with $\overset{\cdot }{V}$O_2_ when compared to other available commercial devices [11]. Possibly by generating a running power metric based on estimates of horizontal velocity and vertical displacement using acceleration data, it is purported that Stryd power can be used as a proxy for metabolic PO, despite changes in external conditions such as air resistance or gradient [13]. Thus, Stryd running power can theoretically quantify training intensity in a manner analogous to cycling mechanical PO and could be superior to conventional measurement approaches using running speed. Despite evidence of repeatability [11], reliability [14,15], stability during prolonged running [16], and strong linear correlations with running speed [17,18], limited research has investigated the Stryd running metric at stable metabolic work rates relative to exercising thresholds. Thus, to determine the utility of Stryd power to indicate relative exercise intensity and assess running fitness and performance, the relationship between Stryd mechanical power and metabolic power needs to be established using an exercise intensity domain training approach (i.e., evaluating running power metrics during steady-state exercise relative to the gas exchange threshold (GET) and maximal metabolic steady state (MMSS)).

Prior to determining whether Stryd running power can monitor training, like cycling power output, in uncontrolled environments (e.g., variable inclines, wind speeds, and surfaces), the primary purpose of the present study was to evaluate the Stryd power metric in a controlled environment (i.e., in situ). Using an exercise intensity domain approach, we assessed the stability, sensitivity, and reliability of Stryd at stable metabolic work rates to (i) determine the efficacy of Stryd running power as a training intensity and running performance metric, (ii) explore the relationship between running power and running economy (RE), (iii) estimate mechanical efficiency during constant-speed treadmill running, and (iv) contrast steady-state measurements with measurements derived from incremental exercise. We hypothesized that Stryd running power would be repeatable across two visits, stable during a 30-min run, and sensitive to running speeds near the maximal lactate steady state (MLSS)—a proxy measure of the MMSS. In addition, we hypothesized that Stryd power would be strongly associated with running speed, $\overset{\cdot }{V}$O_2_, and RE measurements, thereby providing a tool to guide exercise training and assess running fitness.

## 2. Materials and Methods

### 2.1. Participants

Fifteen (8 male; 7 female) recreationally active or trained/developmental runners [19] (mean [SD]; age = 30.2 [4.3] years; body mass = 68.8 [8.2] kg; height = 173.2 [8.4] cm; $\overset{\cdot }{V}$O_2_max, 54.5 [6.5] ml·kg^−^^1^·min^−^^1^) were recruited using convenience sampling. Participants were included if they were healthy, uninjured, and between 18 and 45 years of age, with recent 10-km performances of ≤50 min and ≤55 min for males and females, respectively. Within the 3 months prior to testing, runners reported exercising an average of 3.5 [1.4] days per week, running an average of 27.7 [17.1] km each week, and having 10-km best performance times of 44.6 [6.5] min. Written informed consent was provided by the runners to participate in the experimental procedures, which were approved by the University of Calgary Conjoint Health Research Ethics Board (REB20-0111) and conducted in accordance with the declaration of Helsinki, except for pre-trial registration. Participants had the option to cease participation at any time during the experimental procedures. Prior to test administration, runners completed the physical activity readiness questionnaire (PAR-Q+) to identify contraindications to exercise testing and to ensure that participants were free of medical conditions and injuries that could interfere with metabolic and cardiorespiratory exercise responses. All runners provided their own lightweight running shoes and wore the same shoes for all testing sessions.

### 2.2. Experimental Design

Runners visited the laboratory for five to six exercise testing sessions, with a minimum of 48 h between visits. The exercise sessions included: (1) a “Step-Ramp-Step” (SRS) exercise test to determine maximal exercising parameters [20]; (2) a series of 3–4 constant-speed bouts to determine the MLSS; and (3) a repeated trial at the MLSS running speed. Runners were asked to refrain from smoking, eating, or consuming caffeine within 2 h prior to their testing sessions. Runners did not engage in strenuous exercise on the same day as the testing sessions. A manuscript validating the SRS approach to identify the running speed and Stryd running power associated with the MLSS has been published [20]; however, despite the overlap in experimental procedures, the results presented herein are distinct.

### 2.3. Exercise Protocols

#### 2.3.1. Step-Ramp-Step (SRS) Protocol

As described in detail in our previous study [20], runners performed an SRS exercise protocol during their first testing visit to establish their maximal exercising values and estimate the running speed associated with the MLSS. This SRS protocol was modified for treadmill running from a cycle ergometer-based method [21]. Of relevance to the present study, the SRS protocol involved a moderate-intensity step-transition (MOD; 6 min at 1.9 m·s^−^^1^, 6 min at 2.4 m·s^−^^1^, and 6 min at 1.9 m·s^−^^1^); an incremental treadmill running test (an initial speed of 1.9 m·s^−^^1^, increasing by ~0.2 m·s^−^^1^ (i.e., 0.5 mph) per min, until volitional exhaustion); and a heavy-intensity step transition (HVY; 4 min of treadmill running at 1.9 m·s^−^^1^, followed by 12 min of treadmill running at a speed associated with the heavy-intensity exercise domain). The incremental treadmill test immediately preceded the MOD step, but the participants recovered passively for 30 min between the incremental test and the HVY step. The SRS protocol facilitated the identification of the MLSS in 2–3 constant-speed trials [20].

#### 2.3.2. Constant-Speed Treadmill Running—MLSS Determination

Following the initial SRS testing visit, runners completed the constant-speed exercise sessions during 4 to 5 separate visits to the lab. These visits consisted of 5 min of treadmill running at 1.9 m·s^−^^1^, followed by treadmill running at the predetermined testing speed. During all constant-speed testing visits, participants were encouraged to run until volitional exhaustion, up to a maximum duration of 45 min (excluding warm-up). Data collected beyond 30 min were not used in this study. All runners performed their initial constant-speed test at the running speed estimated to be the MLSS by the SRS protocol. Depending on whether the difference between the 10- and 30-min blood lactate concentrations ([BLa]) was ≤1 mmol·L^−^^1^ or >1 mmol·L^−^^1^, the subsequent visit was performed at a treadmill speed that was 5% faster or 5% slower, respectively. The MLSS for each runner was identified as the highest treadmill speed whereby at least 30 min of exercise was performed and the difference between the [Bla] at 10 and 30 min was ≤1 mmol·L^−^^1^ [22]. All participants performed constant-speed treadmill running trials at the MLSS, 5% above the MLSS, 5% below the MLSS, and once more at the MLSS. Data analysis was primarily based on data collected at the 10- and 30-min (or at task failure if <30 min) time points.

### 2.4. Equipment and Measurements

#### 2.4.1. Cardiorespiratory Measurements

All exercise sessions were performed on a treadmill (Desmo Pro Evo, Woodway USA Inc., Waukesha, WI, USA) with an incline set to a 1% gradient [23]. Adjustments to treadmill speed could be made in 0.1 mph increments (i.e., ~0.04 m·s^−^^1^); however, all running speed data were reported in SI units (i.e., m·s^−^^1^). Ventilatory and gas exchange variables were measured using the Quark CPET metabolic cart (COSMED, Rome, Italy), with a mixing chamber (COSMED), facemask (7450 Series V2, Hans-Rudolph, Shawnee, KS, USA), 2-way non-rebreathing valve (Hans-Rudolph), and gas collection hose. The metabolic cart system was calibrated using a 3 L syringe and gas mixture of known composition (5% CO_2_, 16% O_2_, and N_2_ for the balance) prior to each testing visit. For the analysis, 10-s average ventilatory and gas exchange data were used. Heart rates were recorded during all testing sessions using a Polar H10 chest strap (Polar Electro Oy, Kempele, Finland). 

The $\overset{\cdot }{V}$O_2_ associated with a disproportionate increase in the rate of carbon dioxide production ($\overset{\cdot }{V}$CO_2_) and minute ventilation ($\overset{\cdot }{V}$_E_) relative to the increase in $\overset{\cdot }{V}$O_2_ was used to identify the GET [24]. The $\overset{\cdot }{V}$O_2_ associated with a second disproportionate increase in $\overset{\cdot }{V}$_E_ and a disproportionate increase in $\overset{\cdot }{V}$_E_/$\overset{\cdot }{V}$CO_2_ relative to the increase in $\overset{\cdot }{V}$O_2_ was used to identify the respiratory compensation point (RCP) [24,25]. $\overset{\cdot }{V}$O_2_max was identified as the highest 30-s average $\overset{\cdot }{V}$O_2_ achieved during the incremental test. All incremental tests were considered maximal, based on the observation of a $\overset{\cdot }{V}$O_2_ plateau (defined as a change in $\overset{\cdot }{V}$O_2_ of less than 150 mL·min^−^^1^, despite an increased intensity) or any of the following criteria: maximum HR within 10 bpm of the age-predicted maximal value, a respiratory exchange ratio (RER) greater than 1.15, or [Bla] ≥ 8 mmol·L^−^^1^ upon test completion. 

#### 2.4.2. Blood Lactate Measurements

All [Bla] data were collected using capillary blood drawn from a pinprick of the finger and analyzed for [Bla] using the Biosen C-Line (EKF Diagnostics, Cardiff, Wales; *n* = 7) or Lactate Plus (Nova Biomedical, Waltham, MA, USA; *n* = 8) lactate analyzer. Runners straddled the treadmill (~60–75 s) during [Bla] measurements at 10 and 30 min (or at task failure if <30 min).

#### 2.4.3. Perceptual Responses

After familiarization with the scale, the rating of perceived exertion (RPE) was measured every 5 min during constant-speed running, using the Borg RPE scale (6–20) [26]. 

#### 2.4.4. Running Power—Stryd

Running power measurements were made using the Stryd Summit Running Pod (Stryd, Boulder, CO, USA). The Stryd pod, which is a lightweight (8.0 g) and unobtrusive (4.0 cm in length) wearable sensor (Model v.19, firmware v.2.1.16, software v.4), was affixed to the runner’s left shoe, approximately equidistant between the runner’s malleoli and the shoe’s toe. A unique Stryd user profile was created for each runner that included their respective height and body mass, which was kept constant for all testing sessions. The iPhone Stryd application (Apple Inc., Cupertino, CA, USA) was used to pair the Stryd device and collect the Stryd running power data during the testing sessions. Running power data were sampled at 1 Hz (see Figure 1).

### 2.5. Data Analysis

#### 2.5.1. Cardiorespiratory, Running Speed, and Stryd Running Power Data

The average $\overset{\cdot }{V}$O_2_ and running power, measured between minutes 4 and 6 of the MOD step and between minutes 10 and 12 of the HVY step, were calculated from the SRS test. Maximal aerobic speed (MAS) and maximal aerobic power (MAP) were determined as the running speed associated with the highest completed 1-min stage during the incremental test and the average running power during that stage, respectively. 

Cardiorespiratory and running power data used for analysis from the constant-speed MLSS-determination running trials included the 10- and 30-min $\overset{\cdot }{V}$O_2_, $\overset{\cdot }{V}$CO_2_, RER, $\overset{\cdot }{V}$_E_, HR, and running power measures for running trials 5% below, at, and 5% above MLSS. To align with the timing of [BLa] measurement (i.e., a short pause in running), mean values for $\overset{\cdot }{V}$O_2_, $\overset{\cdot }{V}$CO_2_, RER, $\overset{\cdot }{V}$_E_, HR, and Stryd running power were calculated from the 2 min of data collected prior to the 10-min and the 30-min (or at task failure if <30 min) time points. Although the MLSS is thought to represent the highest intensity at which energy provision is supplied exclusively via oxidative metabolism [27], data collected at 5% above the MLSS were included in the analysis due to the similarly stable $\overset{\cdot }{V}$O_2_ measurements between the 10- and 30-min values across the three intensities (i.e., differences between 10- and 30-min $\overset{\cdot }{V}$O_2_ measures were ~50 mL·min^−^^1^ at intensities of 5% below, at, and 5% above the MLSS)—with similar findings previously being reported [28]—and to provide a more comprehensive dataset for the analyses.

#### 2.5.2. Incremental and Constant-Speed $\overset{\cdot }{V}$O_2_–Power and Speed–Power Gains

A least-squares linear regression was performed to calculate the $\overset{\cdot }{V}$O_2_–power gain (i.e., the slope of the regression equation) for each participant during both incremental and constant-speed exercise trials, measured separately. This method allowed for the calculation of a $\overset{\cdot }{V}$O_2_–power gain mean and standard deviation in which comparisons could be made between the incremental and constant-speed running tests and to contrast the measurements with cycling data [29]. The “incremental $\overset{\cdot }{V}$O_2_–power gain” for each runner was calculated as the slope ((mL·min^−^^1^)·W^−^^1^) of a least-squares linear regression line through the incremental exercise $\overset{\cdot }{V}$O_2_–power response, from the onset of a systemic rise in $\overset{\cdot }{V}$O_2_ until test termination or the onset of a plateau, if detected. The “constant-speed $\overset{\cdot }{V}$O_2_–power gain” for each runner was calculated as the slope ((mL·min^−^^1^)·W^−^^1^) of a least-squares linear regression line for the steady-state $\overset{\cdot }{V}$O_2_ and power data from five constant-speed intensities: the MOD and HVY steps from the SRS-protocol and the constant-speed exercise trials 5% below, at, and 5% above the MLSS. Replacing $\overset{\cdot }{V}$O_2_ with running speed, these same procedures were used to calculate the “incremental speed–power gain” and the “constant-speed speed–power gain” for each participant.

#### 2.5.3. Metabolic Power, Mechanical Power, and Mechanical Efficiency

Running metabolic power, mechanical power, and mechanical efficiency measurements were calculated during constant-speed running trials at MOD, HVY, 5% below, at, and 5% above MLSS. Metabolic power was calculated as a gross energy cost per unit of body mass and distance travelled (kJ·kg^−^^1^·km^−^^1^) using $\overset{\cdot }{V}$O_2_ and RER [30]. This calculation of metabolic power was used to represent the energy cost of running (i.e., RE) at each respective intensity, providing a reference measure of RE in which all subsequent comparisons were made. Metabolic power (i.e., Stryd_MET_) was also calculated by expressing the energy cost—using $\overset{\cdot }{V}$O_2_ and RER [30]—per unit of absolute Stryd power ((J·s^−^^1^)·W^−^^1^) and per unit of relative Stryd power ((kJ·s^−^^1^)·(W·kg^−^^1^)^−^^1^). Stryd_MET_ was calculated in isolation from running speed to provide a metric that characterized the metabolic power requirements per unit of Stryd running power. The units used to describe Stryd_MET_ were not simplified in order to distinguish among related terms and to provide units that clearly described the energy cost of running per unit of absolute and relative Stryd running power. Mechanical power (i.e., Stryd_MECH_) was calculated in isolation from the $\overset{\cdot }{V}$O_2_ and RER by expressing Stryd running power (W) in units of J·s^−^^1^ and by converting mechanical power to an absolute energy cost per unit of the distance travelled (kJ·km^−^^1^) and a relative energy cost per unit of distance (kJ·kg^−^^1^·km^−^^1^). Mechanical efficiency (EFF) was calculated as the ratio between Stryd_MECH_ (kJ·kg^−^^1^·km^−^^1^) and metabolic power (kJ·kg^−^^1^·km^−^^1^), expressed as a percentage. 

### 2.6. Statistical Analysis

#### 2.6.1. General

Statistical analyses were performed using the Statistical Package for the Social Sciences (SPSS, version 26, IBM, Armonk, NY, USA). Linear mixed-effects models were performed using the nlme package (version 3.1-157) in RStudio (version 4.2.0) (R Core Team (2018)). Data visualization was performed using Prism (version 9.5.1 for macOS; GraphPad Software, San Diego, CA, USA). Data are presented as mean [standard deviation (SD)]. Statistical significance was set at an α level of <0.05. Where appropriate, Bonferroni post hoc tests were used. Test-retest reliability was measured using two-way mixed effects, absolute agreement, and single-rater intraclass correlation models wherein reliability was interpreted as poor (ICC < 0.5), moderate (0.5 ≤ ICC < 0.75), good (0.75 ≤ ICC < 0.9), or excellent (ICC ≥ 0.9) [31]. 

#### 2.6.2. Stability, Sensitivity, and Reliability

Multiple two-way repeated-measure ANOVAs were used to assess stability (the main effect of duration) and sensitivity (the main effect of intensity) of Stryd running power and the physiological and perceptual responses (i.e., $\overset{\cdot }{V}$O_2_, $\overset{\cdot }{V}$CO_2_, RER, $\overset{\cdot }{V}$_E_, HR, [BLa], and RPE) at 10 min and 30 min (or task failure, if <30 min) during constant-speed treadmill running at 5% below, at, and 5% above the MLSS. For the same variables, the two MLSS trials were compared using paired Student’s *t* tests, intraclass correlations, and Bland–Altman analyses (with 95% limits of agreement) to assess reliability at the 30-min timepoint. Stryd running power stability was further assessed by evaluating the linear association and agreement between the 10- and 30-min running power data for the first MLSS trial using a Pearson’s correlation coefficient and Bland–Altman analysis, respectively.

#### 2.6.3. Stryd Running Power—Association with $\overset{\cdot }{V}$O_2_ and Running Speed

Paired Student’s *t* tests were used to compare the mean $\overset{\cdot }{V}$O_2_–power gains and $\overset{\cdot }{V}$O_2_–speed gains between incremental and constant-speed exercise trials. To determine the association between Stryd running power and training intensity, linear mixed-effects models were used to assess the within-individual and between-individual association between running power and $\overset{\cdot }{V}$O_2_ measurements, and between running power and running speed during the MOD, HVY, MLSS −5%, MLSS, and MLSS +5% running trials. Models included fixed-effects models of absolute running power and relative running power while allowing intercepts as random effects for the participants to account for repeated measurements within individuals [32]. Models were estimated using maximum likelihood, model selection was assessed using a chi-squared likelihood ratio test, and model fit was assessed using pseudo-R^2^ [32]. These analyses were performed for absolute (i.e., W) and relative measures of power (i.e., W·kg^−^^1^). The spread of the participants’ intercepts was compared using the $\overset{\cdot }{V}$O_2_ and absolute power and the $\overset{\cdot }{V}$O_2_ and relative power relationships and using the speed and absolute power and speed and relative power relationships, employing the Pitman–Morgan test for the homogeneity of variance of paired samples.

#### 2.6.4. Stryd Running Power—Running Economy and Efficiency

To determine whether Stryd running power provides an indication of RE during constant-speed treadmill running trials at MLSS, Pearson’s correlation coefficients were calculated between metabolic power (kJ·kg^−^^1^·km^−^^1^) and each of the following variables: absolute Stryd_MECH_ (kJ·km^−^^1^), relative Stryd_MECH_ (kJ·kg^−^^1^·km^−^^1^), absolute Stryd_MET_ ((J·s^−^^1^)·W^−^^1^), and relative Stryd_MET_ ((kJ·s^−^^1^)·(W·kg^−^^1^)^−^^1^). One-way repeated-measures ANOVAs were used to assess the main effect of running intensity on metabolic power, Stryd_MECH_, and Stryd_MET_ measurements. Using the 30-min timepoint mean (i.e., 28–30 min) data from the two MLSS trials, the reliability of metabolic power, Stryd_MECH_, and Stryd_MET_ were assessed by paired Student’s *t* tests, intraclass correlations, and Bland–Altman analyses (with 95% limits of agreement). One-way repeated-measures ANOVAs were also used to determine whether running intensity affected Stryd-derived assessments of EFF.

#### 2.6.5. Stryd Running Power—Aerobic Fitness

To determine whether Stryd running power provides an indication of an athlete’s aerobic fitness during constant-speed treadmill running, Pearson’s correlation coefficients were calculated for the following pairs of variables: $\overset{\cdot }{V}$O_2_ at MLSS and running power at MLSS, and $\overset{\cdot }{V}$O_2_ at MLSS and running speed at MLSS.

## 3. Results

### 3.1. Participants

Table 1 displays the female and male participant characteristics, incremental exercise testing results, and MLSS testing results. All incremental tests were maximal, and the duration of the incremental test portion of the SRS protocol was 12.1 [2.0] min. The measured $\overset{\cdot }{V}$O_2_ during constant-speed running at MOD and HVY was 91.2 [8.0]% and 92.9 [5.6]% of the $\overset{\cdot }{V}$O_2_ at GET and RCP, respectively. All runners completed at least 30 min of treadmill running at 5% below the MLSS, at the MLSS, and during the repeat trial at the MLSS; however, seven runners were unable to complete 30 min of running at 5% above the MLSS.

### 3.2. Stability, Sensitivity, and Reliability of Stryd Running Power

The 10- and 30-min running power measurements taken during constant-speed running trials near the MLSS are presented in Figure 2. While the intensity × duration interaction and the main effect of duration were not statistically significant for running power, there was a significant main effect for running intensity, with significant differences between all pairs of intensities (*p* < 0.001 for all post hoc comparisons; Table 2). 

The 10- and 30-min running power measurements during the repeat constant-speed running trial at MLSS are reported in Table 3. There was excellent reliability and low bias between the running power measured at two time points within one run at the MLSS and across two runs at the MLSS, without differences between repeated trials at the MLSS (Figure 2; Table 3).

### 3.3. Physiological and Perceptual Responses

The duration × intensity interaction was not significant for $\overset{\cdot }{V}$O_2_; however, there was a main effect of intensity, with significant differences across the three running speeds and a main effect of duration, demonstrating 30-min values greater than the 10-min values (*p* < 0.05 for all post hoc comparisons; Table 2). The $\overset{\cdot }{V}$O_2_ values measured at two time points within one run at the MLSS had excellent reliability and low bias across two runs at the MLSS, without differences between repeated trials at the MLSS (Table 3).

Descriptive data and statistical results for $\overset{\cdot }{V}$CO_2_, RER, $\overset{\cdot }{V}$_E_, HR, [BLa], and RPE measured at two time points (10-min and 30-min) for three speeds near the MLSS are reported in the Appendix A.

### 3.4. Stryd Running Power—Association with $\overset{\cdot }{V}$O_2_ and Running Speed

The incremental and constant-speed $\overset{\cdot }{V}$O_2_–power gains and speed–power gains are reported in Table 5. From the constant-speed running trials, the linear mixed-effects models revealed a strong, positive relationship between absolute running power and $\overset{\cdot }{V}$O_2_ and between relative power and $\overset{\cdot }{V}$O_2_ (Table 6; Figure 3). There was significant variance between participant intercepts for both models that differed between models (Table 6; Figure 3), providing evidence that the relationship between absolute power and $\overset{\cdot }{V}$O_2_ was stronger and less variable between the participants than the relationship between relative power and $\overset{\cdot }{V}$O_2_.

Results were similar when speed was used in place of $\overset{\cdot }{V}$O_2_; however, the difference in model intercept variances was in the opposite direction, with a stronger and less variable relationship between relative power and speed compared to absolute power and speed (Table 6; Figure 3).

### 3.5. Stryd Running Power—Association with Running Economy

Based on the constant-speed running trials at MOD and near the MLSS, there were significant effects of intensity on metabolic power, Stryd_MECH_, and Stryd_MET_ measurements (Table 6). The metabolic power and Stryd_MET_ measurements were significantly lower at MOD compared to measurements 5% below, at, and 5% above the MLSS (*p* < 0.001 for all pairwise comparisons; Table 6). In contrast, the Stryd_MECH_ measurements were significantly higher at MOD compared to the three higher intensities (*p* < 0.001 for all comparisons; Table 6). All variables had excellent reliability for the repeated trials at the MLSS, without significant differences between trials at the MLSS (Table 3).

Figure 4 depicts the relationships between metabolic power (kJ·kg^−1^·km^−1^) and absolute Stryd_MECH_ (kJ·km^−1^), relative Stryd_MECH_ (kJ·kg^−1^·km^−1^), absolute Stryd_MET_ ((J·s^−1^)·W^−1^), and relative Stryd_MET_ ((kJ·s^−1^)·(W·kg^−1^)^−1^) at the MLSS. Metabolic power (kJ·kg^−1^·km^−1^) was not significantly correlated with absolute Stryd_MECH_ (kJ·km^−1^) or relative Stryd_MECH_ (kJ·kg^−1^·km^−1^); however, strong positive and moderately positive correlations were detected between metabolic power (kJ·kg^−1^·km^−1^) and absolute Stryd_MET_ (J·s^−1^)·W^−1^) and relative Stryd_MET_ ((kJ·s^−1^)·(W·kg^−1^)^−1^), respectively (Figure 4). The results for other intensities were similar.

### 3.6. Stryd Running Power—Estimates of Mechanical Running Efficiency

There was a statistically significant main effect of running speed for EFF (*p* < 0.001; Figure 5). Pairwise comparisons revealed that EFF was significantly higher at MOD (25.0 [1.8]%) compared to HVY (21.3 [1.2]%), 5% below MLSS (20.9 [1.7]%), MLSS (20.7 [1.4]%), and 5% above MLSS (20.4 [1.4]%) (*p* < 0.001 for these pairwise comparisons). No other significant differences were detected between the EFF measurements.

### 3.7. Stryd Running Power—Association with Aerobic Fitness 

Absolute running power at the MLSS was strongly correlated with absolute $\overset{\cdot }{V}$O_2_ at the MLSS, moderately correlated with relative $\overset{\cdot }{V}$O_2_ and absolute running speed at the MLSS, and not correlated with relative running speed at the MLSS (Table 1; Figure 6). Relative running power at the MLSS was not correlated with absolute $\overset{\cdot }{V}$O_2_ at the MLSS, but it was strongly correlated with relative $\overset{\cdot }{V}$O_2_ and absolute running speed at the MLSS and moderately correlated with relative running speed at the MLSS (Table 1; Figure 6).

## 4. Discussion

The results from this investigation support the use of Stryd in research and applied settings. The Stryd running power metric was stable during 30-min constant-speed running trials, repeatable across trials at the MLSS, and sensitive enough to differentiate between trials performed at running speeds of 5% below, at, and 5% above the MLSS threshold. Running power was strongly correlated with running speed and $\overset{\cdot }{V}$O_2_ during constant-speed exercise relative to the GET and MLSS, supporting its use as a training intensity metric. Furthermore, running power measurements at the MLSS were strongly associated with both the $\overset{\cdot }{V}$O_2_ and running speed at the MLSS. Although metabolic power was strongly associated with absolute Stryd_MET_, it appears that Stryd power cannot provide an indication of RE in isolation from metabolic data, as the associations between metabolic power and Stryd_MECH_ were weak. Despite this finding, the mechanical running efficiency derived using Stryd (i.e., EFF) remained consistent and proportional at various exercise intensities near the MLSS threshold.

### 4.1. Stability, Sensitivity, and Reliability

Mean running power measurements were similar across the 10- and 30-min timepoints during constant-speed running trials at 5% below, at, and 5% above the MLSS. Along with a strong correlation, zero bias, and narrow LOA between time points, these findings indicate that the Stryd signal remained stable during constant-speed treadmill running. Running power across two runs at the MLSS was also strongly correlated, with a near-zero bias and narrow LOA, indicating the excellent day-to-day reliability of the metric. Furthermore, the Stryd power metric was able to distinguish between exercise intensities near the MLSS. In agreement with our results, previous investigations also reported that Stryd running power was stable during constant-speed running [33], repeatable [11], and sensitive between conditions [34]; however, our investigation is the first to evaluate these running power parameters near the MLSS, an important threshold for training programs and fitness assessment [35,36]. In support of the Stryd running power metric results, besides a significantly lower RPE measurement during the second compared to the first MLSS trial (i.e., 0.8 units on the Borg 6–20 scale), which may indicate increased comfort during testing, the $\overset{\cdot }{V}$O_2_ (Table 2) and other physiological and perceptual responses to running near the MLSS were also stable, sensitive, and reliable (Appendix A).

### 4.2. Stryd Running Power and Exercise Intensity

The strong associations observed between running power, $\overset{\cdot }{V}$O_2_, and speed support the use of Stryd running power to guide exercise training relative to the exercise intensity domains. Of note, the relationship between Stryd running power and $\overset{\cdot }{V}$O_2_, considered at the group level, was stronger when running power was expressed in absolute units, whereas the relationship between Stryd running power and speed, at the group level, was stronger when running power was expressed in relative units. In practice, our results suggest that absolute Stryd power may be best used as a metric to approximate the rate of absolute oxygen consumption, while relative running power may be best used to indicate running speed—at least during treadmill running. Due to the varying methodological approaches used to establish $\overset{\cdot }{V}$O_2_–power relationships in previous research [11,17,18,37,38,39], it is difficult to make comparisons across studies.

A major strength of the current investigation is that exercise intensity domains were delineated and the $\overset{\cdot }{V}$O_2_ was subsequently evaluated during appropriate durations of constant-speed running before examining the relationship between $\overset{\cdot }{V}$O_2_ and running power. Exercise in the heavy-intensity domain can result in a slow $\overset{\cdot }{V}$O_2_ component that delays the attainment of a steady $\overset{\cdot }{V}$O_2_ measure by ~10–15 min or longer [40]. Thus, without appropriately delineating the exercise intensity domain, it is difficult to discern whether a given absolute work rate or stage duration will produce steady-state exercising conditions. The influence that intensity domain and $\overset{\cdot }{V}$O_2_ kinetic responses have on subsequent $\overset{\cdot }{V}$O_2_–power relationships can be highlighted by the substantial difference between the incremental (i.e., 11.6 [1.5] (mL·min^−^^1^)·W^−^^1^) and constant-speed $\overset{\cdot }{V}$O_2_–power gains (i.e., 19.8 [3.5] (mL·min^−^^1^)·W^−^^1^). Of interest, this $\overset{\cdot }{V}$O_2_–power gain from incremental treadmill running, measured using Stryd running power, is similar to previously observed $\overset{\cdot }{V}$O_2_–power gain measured during 15 W·min^−^^1^ incremental cycling protocols (i.e., 11.3 [1.2] (mL·min^−^^1^)·W^−^^1^) [29]. 

### 4.3. Stryd Running Power and Running Fitness

Runners with greater MLSS running powers displayed greater $\overset{\cdot }{V}$O_2_ and running speeds at the MLSS (Figure 6). As the $\overset{\cdot }{V}$O_2_ and running speed associated with the MLSS are strong predictors of running performance [41,42], at least in samples with broad aerobic fitness ranges, it appears that the Stryd running power metric can be used to indicate fitness in a similar manner to that used for cycling PO from constant-intensity exercise [43,44]; however, in contrast to cycling, where the cycling speed at any given PO is primarily dictated by surface area and aerodynamics [45], body mass has a more substantial influence on the relationship between Stryd running power and running speed. Thus, while absolute Stryd running power may be used to estimate fitness in terms of absolute $\overset{\cdot }{V}$O_2_ at the MLSS, in order to evaluate fitness from a speed perspective, it is best to interpret Stryd running power relative to body mass or to only interpret the speed–power relationship relative to the individual. Previous investigations have also reported strong associations between Stryd assessments of critical power (CP) and fitness metrics such as the RCP and $\overset{\cdot }{V}$O_2_max [38,46], providing further support for the utility of Stryd to quantify running fitness.

### 4.4. Stryd Running Power, Running Economy, and Mechanical Efficiency

Although strong associations were observed between absolute running power and $\overset{\cdot }{V}$O_2_ during constant-speed treadmill running conditions, there was a degree of variability between the measured $\overset{\cdot }{V}$O_2_ for a given absolute running power (Figure 3). A large proportion of this variance may be explained by the range of Stryd_MET_ requirements for a specific metabolic power between runners (Figure 4). Indeed, runners with greater absolute Stryd_MET_ measurements also exhibited greater metabolic power measures during each constant-speed running intensity test (i.e., MOD and near to the MLSS). Although this finding may suggest that Stryd can be used as an indication of RE, the strong relationship between running power and running speed likely explains this finding. Accordingly, when examining the relationship between metabolic power and Stryd_MECH_ (i.e., determining whether the Stryd running power metric can be used in isolation from energy expenditure to approximate the RE), there is no indication that Stryd_MECH_ is related to RE (i.e., metabolic power), suggesting that this approach cannot distinguish between more and less economical runners. Previous investigations have similarly concluded that Stryd running power metrics may be insufficient for detecting differences in RE between trained runners [46,47] or detecting worsened RE (i.e., increased $\overset{\cdot }{V}$O_2_ at a given running speed) after purposefully altering running biomechanics [37]. 

Our Stryd-derived measures of mechanical efficiency (~21–25%) are lower than previous estimates of “apparent” running mechanical efficiency during level running (e.g., ~50–70%) [6,7,39] but are similar to estimates of gross cycling efficiency (e.g., ~20–25%) [4]. Furthermore, in comparison with the up to ~20% difference in previously reported estimates of running efficiency measurements at various running speeds [6,7,39], the Stryd estimates of running mechanical efficiency for level running during MOD and heavy-intensity running were relatively small (i.e., ~4%). Consequently, our results indicate that Stryd-based measures of mechanical running efficiency remain relatively stable at various submaximal intensities and that the metabolic requirement per unit of Stryd running power and the metabolic requirement per unit of cycling PO are similar.

Despite certain limitations related to the accurate detection of changing RE [37,47] and the quantification of running mechanical PO [39], our data suggest that foot-worn running power metrics can still be used to monitor training and quantify running performance. Although these findings do question the ability of Stryd running power to accurately represent the running mechanical PO, we suggest that a wearable running power device need not evaluate running power in a manner that is true to the definition of mechanical PO to be useful. Indeed, as the relationship between metabolic demand and measurements of running mechanical PO may vary with running speed, incline, and surface [6,7,8], a running training tool that provides a consistent and seemingly equivalent evaluation of metabolic demand may be more useful than one that evaluates external work rate, particularly for such applied uses.

### 4.5. Experimental Considerations

Several limitations warrant discussion. Firstly, as all testing was performed on a treadmill with a fixed incline (1%), it remains unknown whether our findings can be extended to outdoor running conditions under variable running gradients, surfaces, or air resistances. With varying inclines, Stryd has shown evidence of repeatability [11] and strong correlations with $\overset{\cdot }{V}$O_2_ [11,18,39], but the influence of variable running gradients and surfaces on metabolic cost requires further investigation. Secondly, it remains unknown whether Stryd power can adjust for changes in air resistance, such as changes in wind speed. For example, changes in air resistance (e.g., wind, drafting, or drag) impact the cycling $\overset{\cdot }{V}$O_2_–speed relationship [45] without influencing the $\overset{\cdot }{V}$O_2_–PO relationship. As Stryd seemingly derives its estimate of running power by quantifying positive changes in vertical displacement and horizontal velocities, whether it can account for the increases in mechanical PO required to overcome greater air resistance is unclear [13]. Despite evidence that Stryd may detect changes in wind speed [48] and has introduced a metric, “Air power”, to adjust running power based on changes in air resistance from increasing or decreasing wind speeds and/or running speeds [49], it remains unknown whether the Stryd power metric–$\overset{\cdot }{V}$O_2_ relationship is linear in uncontrolled environments. 

## 5. Conclusions

A wide variety of internal and external load-monitoring methods have been used in endurance sports, such as running speed and pace, RPE, [BLa], HR, step count, step frequency, and distance [50]; however, none of these variables provide a continuous, instantaneous, and reliable method to measure training intensity, and imprecise measurements of training stress may negatively affect performance and elevate injury risk. With evidence of stability, reliability, and sensitivity, our study suggests that Stryd’s foot-worn wearable device can be used to monitor training intensity and quantify aerobic fitness. While the impact of variable running gradients, surfaces, and air resistance on the Stryd running power metric still needs to be assessed, our results support the use of Styrd running power to delineate exercise intensity domains, guide training intensity, and assess aerobic fitness during level treadmill running.

## Figures and Tables

**Figure 1 sensors-23-08729-f001:**
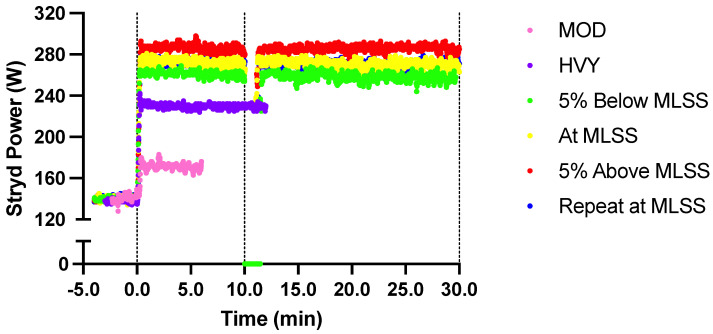
Example of the running power signal during constant-speed treadmill running at different intensities for one participant. Data are shown for the moderate (MOD; 6 min) and heavy (HVY; 12 min) intensity steps, and during 30 min of running at 5% below the maximal lactate steady state (MLSS), at the MLSS, 5% above the MLSS, and during a repeat trial at the MLSS, preceded by running power data recorded for 3–4 min at a running speed of 1.9 m·s^−1^. Running power data were not collected during the first ~1–2 min of each exercise protocol (i.e., warm-up) to allow for synchronization with other measurements. Note that the repeat MLSS trial is obscured by the first MLSS trial.

**Figure 2 sensors-23-08729-f002:**
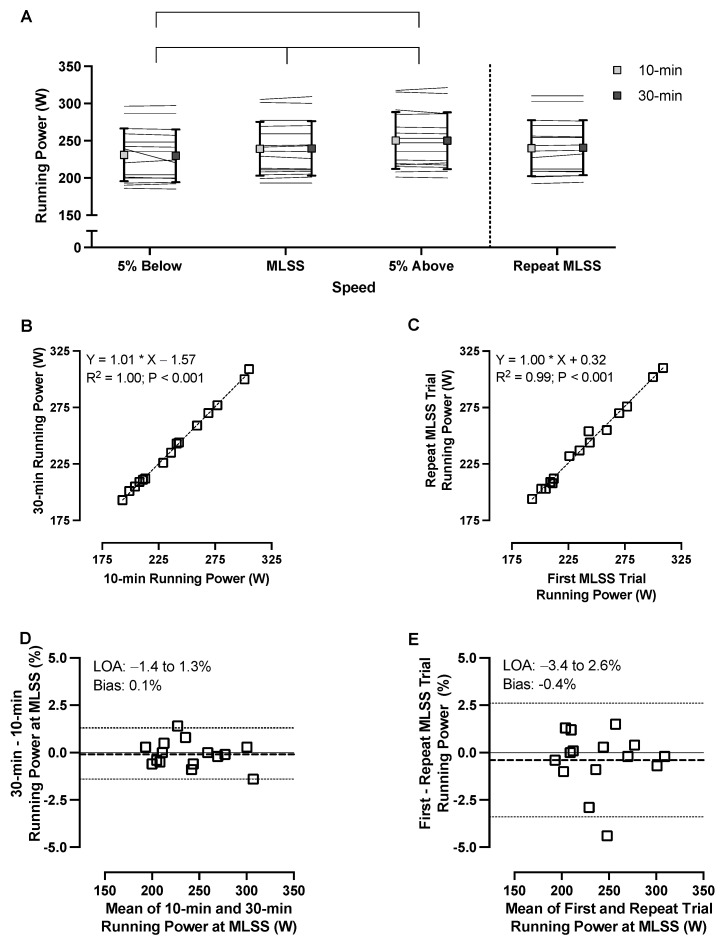
Running power data near the maximal lactate steady state (MLSS). Panel (**A**) shows the comparison between the 10-min and 30-min mean running power measurements during treadmill running near the MLSS. Lines representing individual participants, asterisks (*) indicate statistically significant differences between speeds, and error bars represent one standard deviation. Panel (**B**) shows the relationship between 10 min and 30 min of running power from the first run at the MLSS, and Panel (**C**) shows the relationship between 30 min of running power from the two separate runs at the MLSS. Panels (**D**,**E**) show Bland–Altman plots corresponding to the data in Panels **B** and **C**, respectively. In Panels (**B**–**E**), squares represent individual data, solid lines represent y=0, dashed lines represent bias, and dotted lines represent 95% limits of agreement. *n* = 15 for all panels.

**Figure 3 sensors-23-08729-f003:**
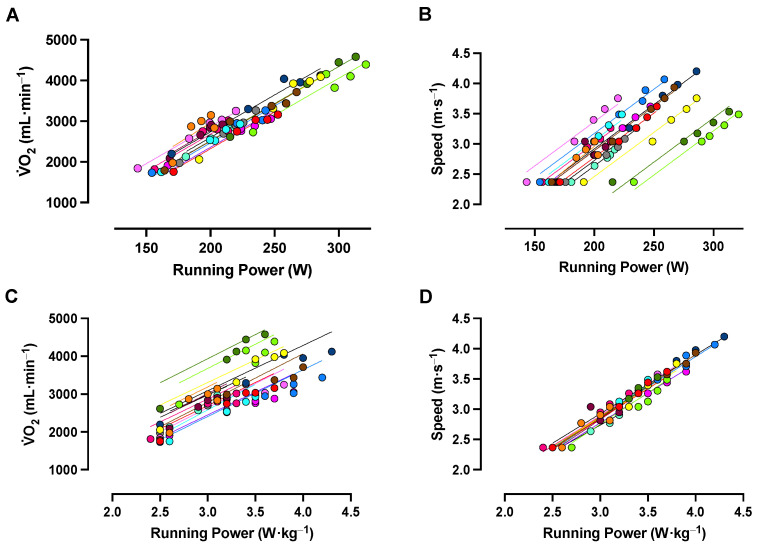
Relationships between absolute and relative running power, running speed, and oxygen uptake ($\overset{\cdot }{V}$O_2_). Panels (**A**,**B**) show the relationships between absolute running power and $\overset{\cdot }{V}$O_2_ and between absolute running power and running speed for each participant during the moderate (MOD) and heavy (HVY) intensity steps and constant-speed trials near the maximal lactate steady state (MLSS). Panels (**C**,**D**) show the relationships between relative running power and $\overset{\cdot }{V}$O_2_ and between relative running power and running speed for each participant at each running intensity, respectively. Each color represents a single participant’s set of trials. *N* = 15 for all panels.

**Figure 4 sensors-23-08729-f004:**
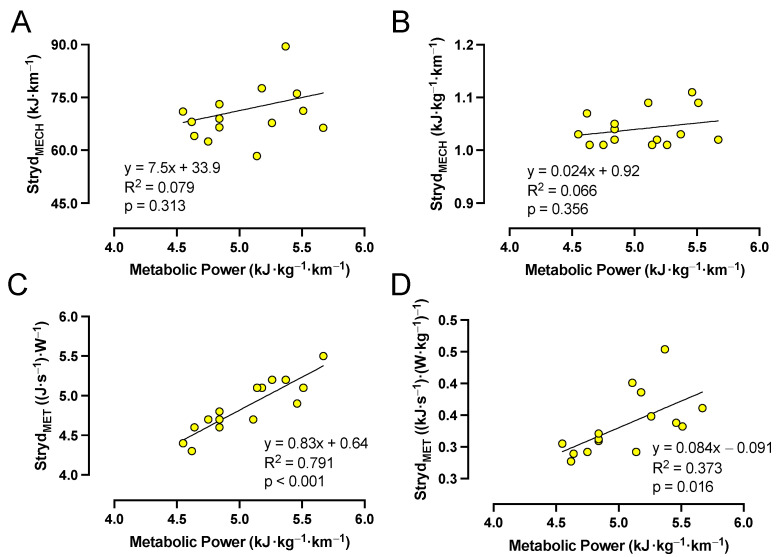
Relationships between metabolic power and absolute Stryd_MECH_ (**A**), relative Stryd_MECH_ (**B**), absolute Stryd_MET_ (**C**), and relative Stryd_MET_ (**D**) during constant-speed running trials performed at the maximal lactate steady state (MLSS). Circles represent individual data. *n* = 15 for all panels.

**Figure 5 sensors-23-08729-f005:**
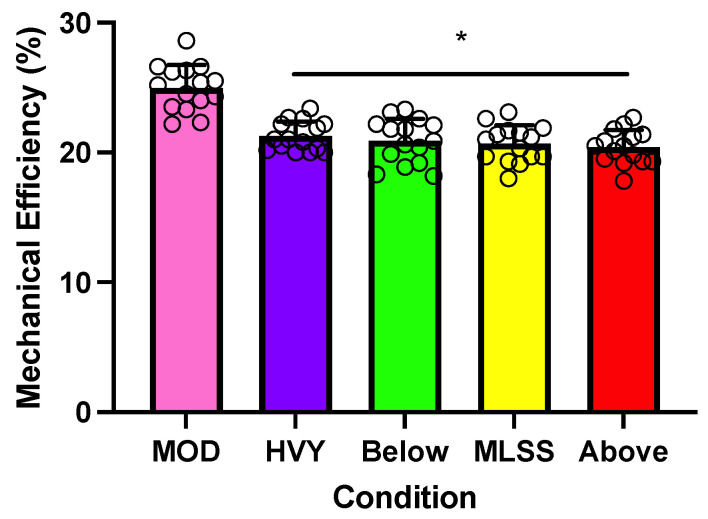
Average running mechanical efficiency (EFF) measurements during the moderate- (MOD) and heavy-intensity (HVY) steps, and during constant-speed trials near the maximal lactate steady state (MLSS). The asterisks (*) indicate statistically significant differences between intensities. Error bars represent one standard deviation. Circles represent individual data. *n* = 15.

**Figure 6 sensors-23-08729-f006:**
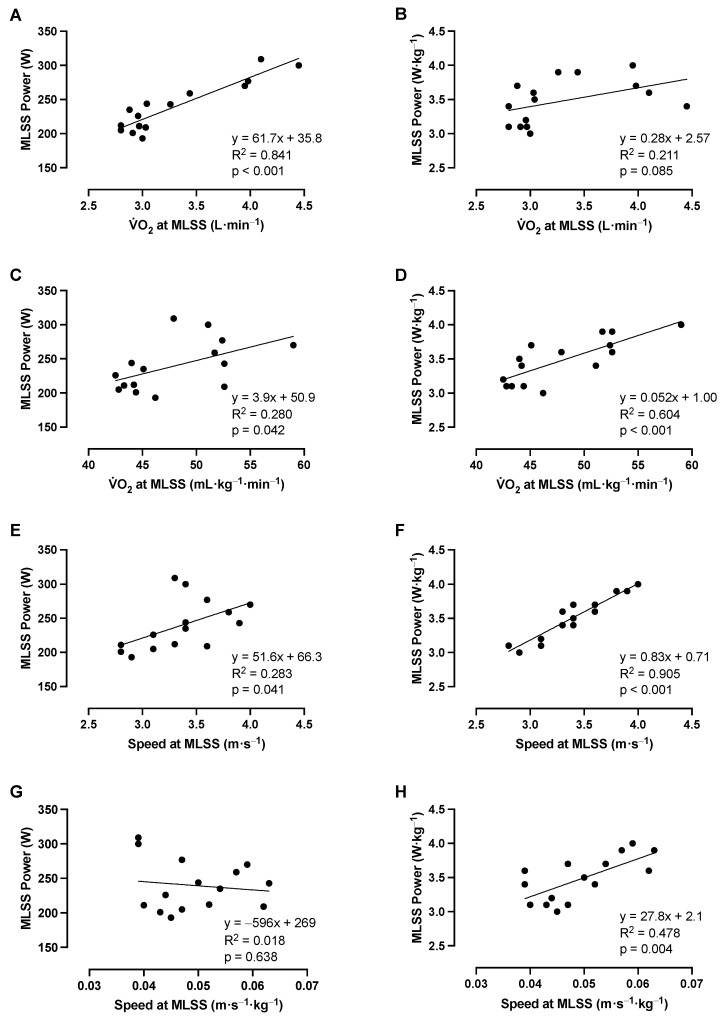
Relationships between absolute and relative running power, running speed, and oxygen uptake ($\overset{\cdot }{V}$O_2_) at the maximal lactate steady state (MLSS). Panels (**A**–**D**) show the relationship between $\overset{\cdot }{V}$O_2_ at the MLSS and running power at the MLSS in absolute and relative units. Panels (**E**–**H**) show the relationship between running speed at the MLSS and running power at the MLSS in absolute and relative units. Individual data are plotted, along with the regression lines. *n* = 15 for all panels.

**Table 1 sensors-23-08729-t001:** Participant characteristics, maximal exercise results, and maximal lactate steady-state (MLSS) results.

Sex (*n*)	Weight (kg)	Maximal Exercising Measurements			MLSS Measurements		
		$\overset{\cdot }{V}$O_2_max (L·min^−1^)	MAS (m·s^−1^)	MAP (W)	$\overset{\cdot }{V}$O_2_ at MLSS (L·min^−1^)	Speed at MLSS (m·s^−1^)	Power at MLSS (W)
Female (7)	65.5 [2.9]	3.37 [0.26]	4.29 [0.44]	280 [19]	2.96 [0.17]	3.29 [0.34]	224 [18]
Male (8)	71.7 [10.4]	4.10 [0.82]	4.54 [0.44]	330 [58]	3.61 [0.58]	3.43 [0.42]	254 [44]
Total (15)	68.8 [8.2]	3.76 [0.71]	4.42 [0.44]	307 [50]	3.30 [0.54]	3.35 [0.37]	240 [37]

$\overset{\cdot }{V}$O_2_max, maximal oxygen uptake; MAS, maximal aerobic speed; MAP, maximal aerobic power. Data are reported as mean [standard deviation].

**Table 2 sensors-23-08729-t002:** $\overset{\cdot }{V}$O_2_ and running power responses to exercise near the maximal lactate steady state (MLSS).

	5% below MLSS		At MLSS		5% above MLSS		ANOVA (DxI, D, I) ^b^
	10 min	30 min	10 min	30 min	10 min	30 min ^a^	
$\overset{\cdot }{V}$O_2_ (L·min^−1^)	3.12 [0.55] ^¶^	3.16 [0.56] *^,¶^	3.26 [0.54] ^†^	3.30 [0.54] *^,†^	3.41 [0.55] ^†,¶^	3.46 [0.57] *^,†,¶^	0.202, 0.001, <0.001
Running Power (W)	231 [35] ^¶^	230 [35] ^¶^	239 [36] ^†^	240 [37] ^†^	250 [38] ^†,¶^	250 [38] ^†,¶^	0.334, 0.528, <0.001

DxI, duration by intensity interaction; D, duration; I, intensity; $\overset{\cdot }{V}$O_2_, oxygen uptake. ^a^ Or the final 2 min if task failure was < 30 min. ^b^ *p*-values are provided for these statistical tests. The * denotes a significant difference from the 10-min timepoint at the same intensity (*p* < 0.05); the ^†^ denotes a significant difference from 5% below the MLSS (*p* < 0.05); the ^¶^ denotes a significant difference from the MLSS (*p* < 0.05). *n* = 15 for all variables. Data are reported as mean [standard deviation].

**Table 3 sensors-23-08729-t003:** Reliability of $\overset{\cdot }{V}$O_2_ and running power responses and metabolic and mechanical power measurements to exercise at the maximal lactate steady state (MLSS).

	At MLSS (Repeat) ^a^		Reliability of Repeated Runs at MLSS (30-min)			
	10 min	30 min	*t* Test ^b^	Bias	LOA	ICC
$\overset{\cdot }{V}$O_2_ (L·min^−1^)	3.25 [0.54]	3.26 [0.52]	0.177	0.04	−0.18 to 0.27	0.99 (0.96 to 1.00)
Running Power (W)	240 [37]	241 [37]	0.322	−1	−8 to 6	1.00 (0.99 to 1.00)
Metabolic Power (kJ·kg^−1^·km^−1^)	-	4.99 [0.29]	0.249	0.06	−0.30 to 0.42	0.91 (0.74 to 0.97)
Stryd_MECH_ (kJ·km^−1^)	-	71.9 [9.5]	0.324	−0.28	−2.32 to 1.77	1.00 (0.99 to 1.00)
Stryd_MECH_ (kJ·kg^−1^·km^−1^)	-	1.04 [0.03]	0.331	0	−0.04 to 0.03	0.94 (0.82 to 0.98)
Stryd_MET_((J·s^−1^)·W^−1^)	-	4.78 [0.30]	0.153	0.07	−0.29 to 0.44	0.91 (0.73 to 0.97)
Stryd_MET_ ((kJ·s^−1^)·(W·kg^−1^)^−1^)	-	0.33 [0.04]	0.121	0.01	−0.02 to 0.03	0.98 (0.93 to 0.99)

LOA, limits of agreement; ICC, intraclass correlation. ^a^ See Table 2 and Table 4 for data from the first MLSS trial. Note that metabolic and mechanical power measures are based on the 30-min time point only. ^b^ *p*-values are provided for these statistical tests. Data are reported as mean [standard deviation].

**Table 4 sensors-23-08729-t004:** Mean metabolic and mechanical power measures during the moderate-intensity step (MOD) and during constant-speed running 5% below, at, and 5% above the maximal lactate steady state (MLSS).

	6-min	30-min			ANOVA (*p*-Value)
	MOD	5% below MLSS	At MLSS	5% above MLSS	
Metabolic Power (kJ·kg^−1^·km^−1^)	4.31 [0.36] *^,†,¶^	5.05 [0.35]	5.05 [0.36]	5.07 [0.30]	<0.001
Stryd_MECH_ (kJ·km^−1^)	73.8 [9.8] *^,†,¶^	72.3 [9.6] *^,¶^	71.7 [9.5]	71.1 [9.0]	<0.001
Stryd_MECH_ (kJ·kg^−1^·km^−1^)	1.07 [0.03] *^,†,¶^	1.05 [0.03] ^¶^	1.04 [0.03]	1.03 [0.03]	<0.001
Stryd_MET_ ((J·s^−1^)·W^−1^)	4.02 [0.29] *^,†,¶^	4.82 [0.40]	4.86 [0.34]	4.91 [0.32]	<0.001
Stryd_MET_ ((kJ·s^−1^)·(W·kg^−1^)^−1^)	0.28 [0.04] *^,†,¶^	0.33 [0.05]	0.33 [0.05]	0.34 [0.05]	<0.001

Data are based on the moderate-intensity step (MOD) from the “Step-Ramp-Step” protocol or the 30-min timepoint of the indicated trial. * Denotes a significant difference between the denoted intensity compared to MLSS (*p* < 0.05). ^†^ Denotes a significant difference between the denoted intensity compared to 5% below MLSS (*p* < 0.05). ^¶^ Denotes a significant difference between the denoted intensity compared to 5% above MLSS (*p* < 0.05). Data are reported as mean [standard deviation]. *n* = 15 for all variables.

**Table 5 sensors-23-08729-t005:** The $\overset{\cdot }{V}$O_2_–power gain and speed–power gain calculated from the incremental exercise test and constant-speed running trials.

Variable	Test		*p*-Value
	Incremental	Constant-Speed	
Absolute $\overset{\cdot }{V}$O_2_–power gain ((mL·min^−1^)·W^−1^)	11.6 [1.5]	19.8 [3.5]	<0.001
Relative $\overset{\cdot }{V}$O_2_–power gain ((mL·min^−1^)·(W·kg^−1^) ^−1^)	810.9 [148.9]	1364.2 [298.7]	<0.001
Absolute speed–power gain ((m·s^−1^)·W^−1^)	0.015 [0.002]	0.015 [0.002]	0.365
Relative speed–power gain ((m·s^−1^)·(W·kg^−1^) ^−1^)	1.05 [0.08]	1.03 [0.07]	0.333

Data are reported as mean [standard deviation]. *N* = 15 for all variables.

**Table 6 sensors-23-08729-t006:** The within-individual and between-individual association between running power and $\overset{\cdot }{V}$O_2_ measurements and between running power and running speed during the MOD, HVY, maximal lactate steady state (MLSS) −5%, MLSS, and MLSS +5% running trials.

Variable	b [95% CI]	Statistics	SD [95% CI]	χ^2^ Statistics	Pitman–Morgan Test
Absolute running power and $\overset{\cdot }{V}$O_2_	18.2 [17.1, 19.3]	t(59) = 32.7; *p* < 0.001; R^2^ = 0.97	196.9 [130.7, 296.8]	χ^2^(4) = −494.8; *p* < 0.001	t(13) = −3.08; *p* = 0.009
Relative running power and $\overset{\cdot }{V}$O_2_	1246.3 [1150.2, 1342.4]	t(59) = 25.6; *p* < 0.001; R^2^ = 0.95	414.0 [285.9, 599.6]	χ^2^(4) = 92.3; *p* < 0.001	
Absolute running power and speed	0.015 [0.014, 0.015]	t(59) = 37.9; *p* < 0.001; R^2^ = 0.97	0.414 [0.288, 0.596]	χ^2^(4) = 125.8; *p* < 0.001	t(13) = 15.72; *p* = 0.002
Relative running power and speed	1.01 [0.96, 1.06]	t(59) = 42.7; *p* < 0.001; R^2^ = 0.97	0.063 [0.039, 0.104]	χ^2^ (4) = 12.7; *p* < 0.001	

b denotes the calculated slope from the linear mixed-effects model. The units of the slope are (mL·min^−1^)·W^−1^ and (mL·min^−1^)·W^−1^ for absolute and relative running power and $\overset{\cdot }{V}$O_2_, respectively, and (m·s^−1^)·W^−1^ and (m·s^−1^)·(W·kg^−1^)^−1^ for absolute and relative running power and speed, respectively. SD denotes the standard deviation, which is presented in units of mL·min^−1^ and m·s^−1^ for $\overset{\cdot }{V}$O_2_ and speed variables, respectively.

## Data Availability

Individual data are shown where possible; however, other data are available from the corresponding author upon reasonable request.

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
