# Peer review of "Is Running Power a Useful Metric? Quantifying Training Intensity and Aerobic Fitness Using Stryd Running Power Near the Maximal Lactate Steady State"

_sensors, 2023, doi:10.3390/s23218729_

Round 1
Reviewer 1 Report
Review comments sensors-2662249
Article title: Is Running Power a Useful Metric? Quantifying Training In- 2 tensity and Aerobic Fitness using Stryd Running Power Near 3 the Maximal Lactate Steady State
Note: For specific comments see PDF with annotated comments to address.
General comment’s introduction
The authors provide a good rationale for the usage of the Stryd IMU and the utility for validating it in order for athletes and general population to use it to track exercise regimes. In addition, they provide a clear gap in the literature which they are intent on closing. However, there is a lack of theoretical explanation how the proxy metabolic power output can be equated to that of mechanical power output, and more importantly how this is derived from the Stryd IMU (e.g., algorithms, CNNs etc.).
General comments materials and methods
The method’s section is well detailed and thoroughly described, in addition the authors have extensively described the exercise habits of the participants within the most recent months. However, the structure of some sentences may require a second look to improve readability and there is no mention if the study adhered to the Declaration of Helsinki.
General comments results
Results are well described and features some detailed figures and tables that explain the findings.
General comments discussion
The authors explain the findings and the implications of the study in regard to the utilization of the stryd in documenting exercise intensity. They also mention the difficulties with comparisons between studies due to study heterogeneities. The authors highlight that Stryd based measures of mechanical running efficiency remain relatively stable at various submaximal intensities, but also addresses the difficulty of serve as a proxy for mechanical running power. The authors, however, indicated that quantification of the latter may not be necessary to be useful

Reviewer 2 Report
Dear authors,
congratulations for your work about the utility of Stryd, a commercially available inertial measurement unit, to quantify running intensity and aerobic fitness. You aimed to first evaluate the Stryd power metric in a controlled environment using an exercise intensity domain approach with the stability, sensitivity, and reliability of Stryd at stable metabolic work rate. Second you try to determine the efficacy of Stryd running power as a training intensity and running performance metric; Third you explored the relationship between running power and running economy (RE); and four you try to estimate mechanical efficiency during constant-speed treadmill running.
In general the manuscript is well written and had strong references to support your discussion. Although there are many questions than can be around stryde, i considered that the literature needs this type of manuscripts.
I only have one major suggestion that is to had "a Case study" to your manuscript title and in the manuscript. Your sample is too small and its not representative to take this conclusions.
